# The Differentiable Curry

Martin Abadi   Richard Wei   Gordon Plotkin   Dimitrios Vytiniotis   Dan Belov
Google Brain                           DeepMind

## 1 Introduction

*Differentiable programming* allows programmers to calculate program gradients and unlocks experimentation with new optimizers and neural network architectures. This is why modern deep learning frameworks [1, 23, 20, 7] introduce derivative APIs (e.g. `tf.gradients` in TensorFlow). Programmers ask for the gradient of an objective function with respect to its parameters; which is then used to optimize these parameters, e.g. through stochastic gradient descent.

Recent projects, as Swift for TensorFlow (S4TF) (`www.tensorflow.org/swift`) and Julia Zygote [15], in the spirit of the seminal "Lambda the Ultimate Backpropagator" (LTUB) [21], advocate AD as a *first-class* construct in a general-purpose programming language, and aim to take advantage of traditional *compiler optimizations* for efficient code generation.

The idea is to produce statically, for every differentiable function of type $a \to b$, another function returning the result of the original function and a linear back-propagator map, of type $a \to (b, \text{Tan } b \multimap \text{Tan } a)$. We call these compiler-generated functions the *representation functions* (rep-functions for short) of differentiable functions, and will use $a \rightsquigarrow b$ as a type abbreviation for $a \to (b, \text{Tan } b \multimap \text{Tan } a)$. To achieve this, the AD pass merely composes rep-functions out of primitive rep-functions like those for $(+)$ and $(-)$, by systematically lifting these primitives through the constructs of the programming language.

An important challenge in this setting is the differentiation of functions that accept or return other functions, perhaps capturing (differentiable or non-differentiable) variables. Partial applications must not "forget" to back-propagate to captured variables, and more generally we need AD that provably preserves equational reasoning – needed to justify inlining, common sub-expression elimination etc. As we will see (Section 2), higher-order functions are ubiquitous in modern statically-typed languages, *even inside the implementation of end-to-end first-order programs*. They have to be tackled heads-on to avoid additional complications in a compiler, such as extra inlining and loop unrolling or early defunctionalization, and to allow for separate compilation, to name a few.

This is the challenge we address. We focus on (i) statically-typed, (ii) compile-time, (ii) reverse-mode AD, a scenario exemplified by Swift AD. (`http://bit.ly/swift-autodiff`) Our contributions are:

- Following recent work [11] we introduce combinators for rep-functions that also return *pullback* linear maps (back-propagators), and show how they can be used for AD. Generalizing to higher-order functions boils down to introducing *differentiable* `curry` and `eval` combinators.

- Higher-order functions (like `curry`) accept or return closures. For a function from tensors to tensors, its pullback at some input is also a linear map from tensors to tensors. We use the term *tangent space*, denoted as `Tan t`, for the space of permutations of values of a type `t`. Hence the tangent space of a tensor is just a tensor. But what should be the tangent space of a function type? Perhaps surprisingly, a function type itself is not the right answer. We provide two possible implementations for function tangents and differentiable currying, and explain the tradeoffs.

- The first implementation – novel to our knowledge – is typeable in a simply-typed discipline and bears similarities to tracing but may involve re-computation of the function we differentiate during the execution of its backwards derivative.

- The second is a *dependently typed* version of an idea behind "Lambda the Ultimate Backpropagator" [21], itself inspired by the view of functions as closures whose tangents are the tangents of their environments. We put that idea in the combinatory setting of Elliott. Our work implies that an "erased" version that relies on reinterpret casts in a weaker type system is in fact safe.

## 2 Higher-order functions and AD

Higher-order functions may not seem essential for differentiation but in a general-purpose programming language (e.g. Swift) are actually ubiquitous, even inside the implementation of a first-order program. Consider, for instance, the case of a recurrent neural network (RNN) model. An RNN, in its simplest form, folds a state transformer (called the RNN cell – e.g. a Long Short-Term Memory [14]) through a sequence of inputs and produces a final state. The state is often called the "hidden state" of the RNN:

```
rnnCell :: (Params, Tensor, Tensor) → Tensor
rnnCell (params, hidden_state, input) =
  ...  // return new hidden_state
runRNN :: Params ⤳ Tensor ⤳ [Tensor] ⤳ Tensor
runRNN params init_state xs =
  let g :: Tensor ⤳ Tensor ⤳ Tensor
      g hid input = lstmCell (params, hid, input)
  in fold g init_state xs
```

Here function `g` is the partial application of `rnnCell` on `params`, passed to the recursive function `fold`. This example shows many features required for a general purpose language; (i) partial applications, (ii) recursive functions (such as `fold`), (iii) recursive datatypes (such as [`Tensor`] above). In particular function `g`, the partial application of `rnnCell` captures the parameters and – unlike our simple example above – needs to *back-propagate* to these parameters. Moreover, we cannot eliminate higher-order functions by inlining, unless we unroll `fold`. And even if we could unroll `fold`, `rnnCell` might be imported from a different module whose source is not available for inlining.

In this extended abstract we will only focus on higher-order functions. We will show how we can express differentiation through higher-order programs following Elliott's recipe by providing implementations of a small set of combinators, given below:

```
id :: τ ⤳ τ
curry :: ((τ_a,τ_b) ⤳ τ_c) → (τ_a ⤳ (τ_b ⤳ τ_c))
eval :: (τ ⤳ σ, τ) ⤳ σ
prod :: (τ_y ⤳ τ_a) → (τ_y ⤳ τ_b) → (τ_y ⤳ (τ_a,τ_b))
(∘) :: (τ_a ⤳ τ_b) → (τ_b ⤳ τ_c) → (τ_a ⤳ τ_c)
const_τ :: σ → (τ ⤳ σ)
proj_i :: (τ_1,...,τ_n) ⤳ τ_i
```

Unfolding types, we see that `curry`/`eval` require a definition for function tangents, $\mathtt{Tan}(\tau \leadsto \sigma)$. What these should be (and why) is answered in this work.

$$\boxed{\mathcal{J}[\![\Delta \vdash^{\!\!\!\!\rightharpoonup} e : \tau]\!] = b}$$

$$\frac{b = \mathcal{J}[\![\Delta, (x{:}\tau) \vdash^{\!\!\!\!\rightharpoonup} e : \sigma]\!]}{\mathcal{J}[\![\Delta \vdash^{\!\!\!\!\rightharpoonup} \mathtt{diff}\lambda x{:}\tau.e : \tau \leadsto \sigma]\!] = \mathtt{curry}\ b}\ \text{BDLam}$$

$$\frac{\begin{array}{c} b_1 = \mathcal{J}[\![\Delta \vdash^{\!\!\!\!\rightharpoonup} e_1 : \tau \leadsto \sigma]\!] \\ b_2 = \mathcal{J}[\![\Delta \vdash^{\!\!\!\!\rightharpoonup} e_2 : \tau]\!] \end{array}}{\mathcal{J}[\![\Delta \vdash^{\!\!\!\!\rightharpoonup} e_1\ e_2 : \sigma]\!] = \mathtt{prod}\ b_1\ b_2 \circ \mathtt{eval}}\ \text{BDApp}$$

$$\frac{x\ \text{is}\ i\text{-th variable in}\ \Delta}{\mathcal{J}[\![\Delta \vdash^{\!\!\!\!\rightharpoonup} x : \tau]\!] = \mathtt{proj_i}}\ \text{BDVar}$$

$$\frac{b_1 = \mathcal{J}[\![\Delta \vdash^{\!\!\!\!\rightharpoonup} e_1 : \tau_1]\!] \quad b_2 = \mathcal{J}[\![\Delta \vdash^{\!\!\!\!\rightharpoonup} e_2 : \tau_2]\!]}{\mathcal{J}[\![\Delta \vdash^{\!\!\!\!\rightharpoonup} (e_1, e_2) : (\tau_1, \tau_2)]\!] = \mathtt{prod}\ b_1\ b_2}\ \text{BDProd}$$

Figure 1: Translating to combinators

## 3 Combinatory-style AD

We first illustrate how AD can be implemented using the aforementioned set of combinators.

### 3.1 Step 1: Translate to combinators

We first use the combinator language in the previous section as the target of conversion from a (conventional) call-by-value higher-order lambda calculus of differentiable functions $\lambda_\partial$. The translation, in Figure 1, has (yet) nothing to do with AD; rather it is reminiscent of the well-understood translation of $\lambda$-calculus into *cartesian closed categories* (CCCs) [8]. $\mathcal{J}[\![\Delta \vdash^{\!\!\!\!\rightharpoonup} e : \tau]\!]$ defines such a type-directed translation. The rules ensure that if $\Delta \vdash^{\!\!\!\!\rightharpoonup} e : \tau$ then $\vdash \mathcal{J}[\![\Delta \vdash^{\!\!\!\!\rightharpoonup} e : \tau]\!] : \Delta \leadsto \tau$, where we abuse notation and refer to $\Delta$ as the tuple of all types of environment variables. The conversion also assumes (not shown, for lack of space) that all differentiable primitives, such as $(*) : (\mathtt{Float}, \mathtt{Float}) \leadsto \mathtt{Float}$ come with rep-function implementations.

### 3.2 Step 2: Implement combinators

Next, we implement our combinators, accepting and returning `a ⤳ b` values, i.e. functions of type `a → (b, Tan b → Tan a)`. We also need to define a tangent space `Tan t` for every type `t`. Figure 2 presents such a small library. Figure 6 in the Appendix gives the definition of $\mathcal{T}[\tau]$ for all the types of $\lambda_\partial$. The cases for floats, products, and tensors of floats are standard; also the rules for "discrete" types all return `Unit`. Some combinators (e.g. `prod`) rely on having 0 and (+) defined for every tangent type $\mathcal{T}[\tau]$, also found in Figure 6. In the next sections we proceed to

```
vjp : (τ ⤳ σ) → τ → 𝒯[σ] → 𝒯[τ]
vjp f x gb = snd (f x) gb

mult :: (Float,Float) ⤳ Float
mult (x1,x2) = (x1*x2, λg. (x2*g, x1*g))

proj_left :: (τ,σ) ⤳ τ
proj_left (a,b) = (a, λg. (g, 0))

prod :: (τ ⤳ σ₁) → (τ ⤳ σ₂) → (τ ⤳ (σ₁, σ₂))
prod f g y = let (a, pbf) = f y
                 (b, pbg) = g y
               in ((a, b), λ(ga, gb). pbf ga + pbg gb)
(∘) :: (τₐ ⤳ τ_b) → (τ_b ⤳ τ_c) → (τₐ ⤳ τ_c)
(∘) f g a = let (b, pbf) = f a
                (c, pbg) = g b
              in (c, λgc. pbf (pbg gc))
```

Figure 2: Library of combinators (excerpt)

```
curry :: ((τₐ,τ_b) ⤳ τ_c) → (τₐ ⤳ (τ_b ⤳ τ_c))
curry f = new_f
  where new_f :: τₐ → (τ_b ⤳ τ_c, 𝒯[τ_b ⤳ τ_c] → 𝒯[τₐ])
        new_f t =
          let new_g :: τ_b → (τ_c, 𝒯[τ_c] → 𝒯[τ_b])
              new_g s = let (r, pb) = f(t, s)
                          in (r, λgr. snd (pb gr))
              new_pb :: 𝒯[τ_b ⤳ τ_c] → 𝒯[τₐ]
              new_pb grs =
               let aux (s,gr) = fst (snd (f (t, s)) gr)
               in sum (map aux grs)
          in (new_g, new_pb)
eval :: (τ ⤳ σ, τ) ⤳ σ
eval (f, x) = let (y, pb) = f x
                in (y, λg. ([(x,g)], pb g))
```

Figure 3: Simply-typed differentiable curry/eval

discuss the highlighted parts in Figure 6, to do with function tangents, curry, and eval.

# 4 Simply-typed curry

We define curry and eval in Figure 3. These definitions together with the need for having an addition and a zero operator, effectively *force* the equation $\mathcal{T}[\tau ⤳ \sigma] = [(\tau, \mathcal{T}[\sigma])]$, a *list* of pairs of values and result tangents. Addition and zero are given by list concatenation and the empty list, as we see in the highlighted parts of Figure 6. These lists intuitively track all calls of a function and hence we have to sum up all the resulting tangents from running our func-

tion forwards and then backwards for every element, arriving at the implementation of curry in Figure 3. The eval combinator merely records the primal value (x) and the output tangent (g) in a singleton list.

## 4.1 Properties and metatheory

Is our construction correct? We answer by showing that it respects equational reasoning principles. For example, when given $f : (\tau_1, \tau_2) ⤳ \tau_3$ which we can curry and repeatedly evaluate with an argument of type $\tau_1$ and another of type $\tau_2$, we will get a function that is not only forward-equivalent, but also has an equivalent back-propagator. An example are the forward-equivalent functions foo1 and foo2 in the "Partial Application" column of Figure 4.

Consider foo1 and foo2 in the "Forgetting Results" column of Figure 4. The two functions should be equivalent in forward and reverse mode, but for foo1 we will back-propagate a tangent value of $[(x,0)]$ for the use of g. In foo2, since g is not used at all, we will back-propagate $[]$. We want to therefore treat $[(x,0)]$ and $[]$ as equivalent, even if they are different lists.

Finally, foo1 and foo2 in the "Summing Results" column are forward-equivalent, hence we expect equivalent back-propagators. In the first case, we call f multiple times with the same argument and sum the results; in the second case we call it once. The tangents that are back-propagated to f in the first case will be $[(x,g),(x,g)]$ where $g$ is the tangent corresponding to the result of foo1. In the second case we get $[(x, g + g)]$. We need these two tangents to be treated as equivalent, even if they are different lists.

We have formalized thus a notion of equivalence that goes beyond $\beta$-equivalence for back-propagators, and showed various CCC laws hold of our combinators wrt. that equivalence. These laws guarantee equivalences for the examples presented in this section.

# 5 Dependently-typed curry

Unfortunately the differentiable curry in Figure 3 has a back-propagator new_pb that involves a full *forward* computation of the original function f, at each of the recorded inputs it was applied to – hence suffers from redundant computation. We need something better.

A key insight from the LTUB work [21], leading to an efficient solution, has been this: a back-propagator for a function $\tau ⤳ \sigma$ should take as argument a value of type $\mathcal{T}[\sigma]$ but return not only tangent $\mathcal{T}[\tau]$ but also the tangent of the *environment* $\Delta$ over which the function closed when it was constructed; $\mathcal{T}[\Delta]$. To understand the intuition, it's helpful to think of

| f :: (Float, Float) $\rightsquigarrow$ Float 

 foo1, foo2 :: (Float, Float) $\rightsquigarrow$ Float 
 foo1 (a, b) = ($\lambda$xb → f (a, xb)) b 
 foo2 (a, b) = f (a, b) | foo1, foo2 :: (Float $\rightsquigarrow$ Float, 
 Float $\rightsquigarrow$ Float) 
 $\rightsquigarrow$ Float $\rightsquigarrow$ Float 
 foo1 (f, g) x = fst (f x, g x) 
 foo2 (f, g) x = f x | foo1 :: (Float $\rightsquigarrow$ Float) 
 $\rightsquigarrow$ Float $\rightsquigarrow$ Float 
 foo1 f x = f x + f x 
 foo2 f x = let y = f x in (y + y) |
|---|---|---|
| Partial Application | Forgetting Results | Summing Results |

Figure 4: Example equivalences (`foo1` and `foo2` in each column)

a function value as just (i) a static top-level code pointer that *does not vary with any input*, plus (ii) an environment of captured values that *could vary* as these captured values vary. In other words, for a closure of type $\tau \rightsquigarrow \sigma$, capturing environment $\Delta$, its tangent space is $\mathcal{T}[\tau \rightsquigarrow \sigma] = \mathcal{T}[\Delta]$.

LTUB originally presented an AD system for a dynamically typed language. But in a static type system we immediately hit a problem: it is no longer the case that $\mathcal{T}[\cdot]$ can be a type operator, because different values of type $\tau \rightsquigarrow \sigma$ capture different environments. We therefore revise $\mathcal{T}[\tau]$ to now become a *value-dependent* type operator that takes a value of type $\tau$ as an argument. We write $\mathcal{T}[v : \tau]$ to denote this dependent tangent type, and when the type is obvious from the context we will simply be writing $\mathcal{T}[v]$.

**Definition 5.1** (Dependently typed rep-function). We define $\tau \rightsquigarrow \sigma$ to be the following type:

$$\exists \tau_\Delta^\flat . \Pi(x{:}\tau) . \Sigma(y{:}\sigma) . \mathcal{T}[y] \rightsquigarrow (\mathcal{T}[x], \tau_\Delta^\flat)$$

where $\tau_\Delta^\flat$ denotes a *first-order* type corresponding to the tangents of the closure environment.

**Definition 5.2** (Dependently typed tangents). We define $\mathcal{T}[v : \tau]$ where $v$ is a closed term of type $\tau$ similarly to the previous non-dependent definition, but modify the case for functions as follows:

$$\mathcal{T}[v : \tau \rightsquigarrow \sigma] \quad = \quad \texttt{match } v \texttt{ with exT } \tau_\Delta^\flat {}_- \Rightarrow \tau_\Delta^\flat$$

These definitions – reminiscent of typed closure conversion [19] – deserve some discussion. Definition 5.1 is just a small augmentation of the rep-function type that we have been familiarized with so far. It existentially quantifies over an environment tangent type $\tau_\delta^\flat$, and returns a function that when given an argument $(x{:}\tau)$ it will return a dependent sum $(y{:}\sigma)$ and an additional back-propagator function. The back-propagator also return a $\tau_\Delta^\flat$. Intuitively, is corresponds to the tangent of the environment captured in the function. For this reason Definition 5.2, opens up one of these existential types and returns the witness type.

```
curry :: ((τt,τs) ⤳ τr) → (τt ⤳ (τs ⤳ τr))
curry (exT τᵇ f) = exT () new_f
  where
    new_f :: Π(t:τt). Σ(g : τs ⤳ τr). 𝒯[g] → (τᵇ, 𝒯[t])
    new_f t =
      let gf :: Π(s:τs).Σ(r:τr). 𝒯[r] → ((τᵇ,𝒯[t]), 𝒯[s])
          gf s = let (r, pullback) = f(t,s)
                  in (r, λgr →
                      let (cte,(ctt,cts)) = pullback gr
                      in ((cte,ctt), cts))
          g = exT (τᵇ,𝒯[t]) gf
          new_pb :: 𝒯[g] → (τᵇ,𝒯[t])
          new_pb env = env
      in (g, new_pb)
eval :: (τ ⤳ σ, τ) ⤳ σ
eval = exT () new_f
  where new_f (exT τᵇ f, x) = let (y, pb) = f x
                              in (y, λg → ((), pb g))
comp :: (a ⤳ b) → (b ⤳ c) → (a ⤳ c)
comp (exT τᵇ_f f) (exT τᵇ_g g) = exT (τᵇ_f,τᵇ_g) h
  where h a = let (b, pb_f) = f a
                  (c, pb_g) = g b
              in (c, λgc → let (envg, gb) = pb_g gc
                               (envf, ga) = pb_f gb
                           in ((envf, envg), ga)
```

Figure 5: Dependently-typed differentiable curry/eval

In Figure 5 we give such a curry and eval, in a non-opinionated dependent-type notation (we also have produced in Agda and Coq). The reader is urged to examine the code, but ignore the type signatures, resting assured, it all type checks. Another remark is that composition (also in Figure 5), simply collects together environment tangents. These need not be maps from variables to tangents, rather just tuples.

We have showed that dependently typed currying satisfies similar laws as the simply-typed version. As a final remark, most production languages do not support dependent types. There our solution can be safely implemented using reinterpret casts. This is, in

fact, a tentative proposal for the Swift AD project.[1]

# 6 Discussion and extensions

## 6.1 Internalizing differentiation

We have not yet described how to calculate vector-Jacobian products. Indeed, rep-functions are *ordinary* functions returning linear maps, and our $\lambda_\partial$ of Section 3 does not include ordinary $(\rightarrow)$, $\lambda$, or applications. As a result, it cannot type `vjp` (Figure 2)! We believe there is a principled way to integrate ordinary and differentiable arrows, while still preserving equational reasoning, out of scope for this abstract.

## 6.2 Further features

By taking the same approach of (i) compiling to combinators, and (ii) implementing these combinators in terms of a core language we can show how to introduce control flow and recursion into a differentiable programming language. Supporting richer algebraic types may also be possible, through user-specified definitions for the tangent spaces of these types.

# 7 Related work

We presented the marriage of ideas behind Elliot's categorical presentation of AD [11] and the seminal LTUB work [21]. We extend Elliot's presentation to differentiable currying and evaluation, while putting the ideas of Pearlmutter and Siskind in a typed setting.

The idea of using closures as back-propagators is receiving recent attention. For example Julia Zygote [15] and Swift AD adopts this design. Other recent work follows similar ideas [27, 26] but is using meta-programming as an implementation technique.

There exists work on differentiation semantics [24] and *differentiable categories* [5], usually interpreting types as vector spaces. Related ideas have appeared for higher-order lambda calculi [18, 9] but the underlying foundations only tackle forward mode.

AD has a long history in array programming and scientific computing [13]. Forward-mode AD has been presented before for (first-order) functional programs [16, 10], as libraries in general purpose languages [3], DSLs [6], and more. We urge the reader to consult the comprehensive survey [2]. Recent systems revisit efficient differentiable array programming [22, 25]. There exist also widely used AD libraries for Python [17], and new systems that target

deep learning applications [12] – the latter supporting variable capture through early defunctionalization.

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

# A   Appendix

$$
\begin{aligned}
\mathcal{T}[\texttt{Float}] &= \texttt{Float} \\
\mathcal{T}[\texttt{Tensor(Float)}] &= \texttt{Tensor(Float)} \\
\mathcal{T}[(\tau,\sigma)] &= (\mathcal{T}[\tau], \mathcal{T}[\sigma]) \\
\mathcal{T}[\tau \rightsquigarrow \sigma] &= [(\tau, \mathcal{T}[\sigma])] \\
\mathcal{T}[\texttt{Unit}] &= \texttt{Unit} \\
\mathcal{T}[\texttt{Tensor(Int)}] &= \texttt{Unit} \\
\mathcal{T}[\texttt{Int}] &= \texttt{Unit} \\
\mathcal{T}[\texttt{Bool}] &= \texttt{Unit} \\[6pt]
0_{\mathcal{T}[\texttt{Float}]} &= 0 \\
0_{\mathcal{T}[\texttt{Tensor(Float)}]} &= 0 \\
0_{\mathcal{T}[(\tau,\sigma)]} &= (0_{\mathcal{T}[\tau]}, 0_{\mathcal{T}[\sigma]}) \\
0_{\mathcal{T}[\tau \rightsquigarrow \sigma]} &= [\,] \\
0_{\mathcal{T}[\tau]} &= () \\[6pt]
x_1 +_{\mathcal{T}[\texttt{Float}]} x_2 &= x_1 + x_2 \\
x_1 +_{\mathcal{T}[\texttt{Tensor(Float)}]} x_2 &= x_1 + x_2 \\
(x_{11}, x_{12}) +_{\mathcal{T}[(\tau,\sigma)]} (x_{21}, x_{22}) &= (x_{11} +_{\mathcal{T}[\tau]} x_{12}, \\
&\qquad x_{21} +_{\mathcal{T}[\sigma]} x_{22}) \\
x_1 +_{\mathcal{T}[\tau \rightsquigarrow \sigma]} x_2 &= x_1 \,+\!\!+\, x_2
\end{aligned}
$$

Figure 6: Tangent spaces for $\lambda_\partial$ (simply-typed)

