# OpenReview forum: "The Differentiable Curry"
_NeurIPS.cc/2019/Workshop/Program_Transformations — Program Transformations @NeurIPS2019 Oral_

### Official Review · AnonReviewer2 · 2019-09-29
**Impressive unification of some powerful ideas**

**Confidence:** 4
**Rating:** 10

**Review:**

This paper brings together some threads that have been discussed by FP and AD aficionados for a long time in a formal and general way.  The paper is clearly written, clearly outlining the contributions.

Questions:
1) I didn't understand what is meant by a rep-function in the intro.  First you say " for every differentiable function of type a → b, another function returning the result of the original function and a linear
back-propagator map, of type a → (b, Tan b −◦ Tan a).
We call these compiler-generated functions the representation functions"
and then later
" we introduce combinators for rep-functions that also return pullback linear maps (back-propagators)"
But I thought the rep function already returned pullbacks?

Nitpicks:
- I'm a little sad that 'tensor' has seemingly replaced 'array'.  What is your definition of tensor?
- Is it a bit over-general to say "T [Tensor(Float)] = Tensor(Float)"?  I.e. in the case where these floats represent e.g. a normalized distribution or some other manifold.

---

### Official Review · AnonReviewer1 · 2019-09-30
**Swings and hits the hard problem for reverse mode**

**Confidence:** 5
**Rating:** 9

**Review:**

This is a very nice paper. Giving everyone an intuitive explanation of what the tangent of a function object looks like, both from a theoretical perspective and inside the guts of an implementation, would certainly justify an oral.

My only real question is a minor technical one. I wonder whether the types introduced here would be compatible with the system described in [21], making the differences from that to the approach here in some sense modular, or if the other differences are necessary to get the types to fly.

- I hate the use of the term “tensor” for arrays, but won’t belabour the point.

---

### Public Comment · ~Jeffrey_Mark_Siskind1 · 2019-10-02
**prior work**

Please note that there has been considerable prior work on AD of higher-oder
functions, that has not been cited.

https://docs.lib.purdue.edu/ecetr/367/
https://link.springer.com/chapter/10.1007%2F978-3-540-68942-3_8
https://link.springer.com/article/10.1007/s10990-008-9037-1
https://link.springer.com/chapter/10.1007/978-3-642-30023-3_25
https://www.cambridge.org/core/journals/journal-of-functional-programming/article/perturbation-confusion-in-forward-automatic-differentiation-of-higherorder-functions/A808189A3875A2EDAC6E0D62CF2AD262

---

### Public Comment · ~Dimitrios_Vytiniotis2 · 2019-10-13
**Thanks, and updates**


Dear all,
We have posted minor updates and fixes, plus expanded the related work discussion with (indeed relevant) work mentioned below -- many thanks to everyone for their feedback and help to make this short note better.

The latest manuscript can be found here: https://dimitriv.github.io/papers/hoad-workshop.pdf

---

### Decision · Program_Chairs · 2019-10-01

**Decision:**

Accept (Oral)

**Comment:**

Despite the use of the word "tensor" instead of multi-dimensional array, both reviewers agreed that this is an excellent contribution.